The relationship between characteristics of root morphology and grain filling in wheat under drought stress

Chen Xinyu
Zhu Yu
Ding Yuan
Pan Rumo
Shen Wenyuan
Yu Xurun
Xiong Fei feixiong@yzu.edu.cn
Jiangsu Key Laboratory of Crop Genetics and Physiology/Co-Innovation Center for Modern Production Technology of Grain Crops/Joint International Research Laboratory of Agriculture & Agri-Product Safety of the Ministry of Education, Yangzhou University , Yangzhou , Jiangsu , China
Sun Genlou
Electronic publication date: 2021 Aug 19
Publication date: 2021
Volume: 9
Electronic Location ID: e12015
Received 2021 Jun 4; Accepted 2021 Jul 29
Copyright: ©2021 Chen et al.
Copyright year: 2021
Copyright holder: Chen et al.
License: This is an open access article distributed under the terms of the Creative Commons Attribution License, which permits unrestricted use, distribution, reproduction and adaptation in any medium and for any purpose provided that it is properly attributed. For attribution, the original author(s), title, publication source (PeerJ) and either DOI or URL of the article must be cited.
License URL: https://creativecommons.org/licenses/by/4.0/

Keywords: Wheat, Drought stress, Root, Caryopsis, Correlation

Funding: National Natural Science Foundation of China 31971810 31701351 Priority Academic Program Development of Jiangsu Higher Education Institutions Postgraduate Research & Practice Innovation Program of Jiangsu Province KYCX20_2978 This work was supported by the National Natural Science Foundation of China (No. 31971810, No. 31701351), a Project Funded by the Priority Academic Program Development of Jiangsu Higher Education Institutions (PAPD), the Postgraduate Research & Practice Innovation Program of Jiangsu Province (No. KYCX20_2978). The funders had no role in study design, data collection and analysis, decision to publish, or preparation of the manuscript.

==============================
Drought is a common yield limiting factor in wheat production and has become a significant threat to global food security. Root system is the organ responsible for water uptake from soil and root growth is closely associated with yield and quality of wheat. However, the relationship between morphological and structural characteristics of root growth and caryopsis enrichment in wheat under drought stress is unclear. In this study, two wheat cultivars (YM13 and YN19) were treated with drought from flowering to caryopsis maturity stage. The changes in morphological structure of roots and characteristics of endosperm enrichment were investigated. Drought stress significantly reduced the root length, plant height, root dry weight and aboveground parts dry weight, whereas the root-shoot ratio of YM13 and YN19 increased by 17.65% and 8.33% under drought stress, respectively. The spike length, spike weight, grains number per spike and 1,000-grains weight of mature wheat also significantly declined under drought stress. Meanwhile, the cross section structure of roots was changed with the enlargement of vascular cylinder and dense distribution of xylem vessels under drought stress. Additionally, drought stress affected the substance enrichment in wheat caryopses, decreasing starch accumulation and increasing protein accumulation of endosperm. Correlation analysis suggested that the root length was closely correlated with the relative areas of amyloplast (0.51) and protein body (0.70), and drought stress increased the correlation coefficient (0.79 and 0.78, respectively). While the root dry weight had a significantly positive correlation with the plant height and aboveground parts dry weight. The results can provide theoretical basis for root architecture optimization, water-saving and high-yield cultivation and quality improvement in wheat.

Introduction

Wheat is the most widely distributed cereal crop in the world with the largest planting area, and it plays an important role in global food production, trade and even food security. Drought stress is one of the important abiotic factors that affects crop growth and limits crop yield. Barnabás, Jäger & Fehér (2008) demonstrated that drought stress can affect the yield of cereal crops from morphology, physiology and development. Previous studies showed that drought during wheat flowering and grain filling stages can cause a substantial decrease in yield (Farooq, Hussain & Siddique, 2014; Morales et al., 2020), while Yang & Zhang (2006) found that water deficit during wheat grain filling stage decreased photosynthetic rate and grain filling rate, shortened grain filling time, promoted premature senescence of plants, but increased the metabolism of non-structural carbohydrates from vegetative organs to grains.

Caryopsis is an important organ for storing nutrients in wheat, and its development directly determines the yield and quality of wheat. As the main histological structure, endosperm accounts for more than 85% of the total weight of wheat caryopses. Starch and protein are the main storage substances of endosperm, which accumulate in endosperm cells in the form of amyloplast and protein body respectively (Moore et al., 2016; Seung & Smith, 2019). Previous authors have conducted a lot of researches about the effect of drought stress on the development of wheat caryopses, and found that drought stress caused changes in wheat quality by mainly affecting the accumulation of starch and protein in caryopses (Ahmadi & Baker, 2001; Ge et al., 2012). Lu et al. (2014) pointed out that the contents of total starch and amylopectin, along with the proportion of B-type amyloplast, in wheat endosperm decreased under drought stress. Meanwhile, drought stress increased the protein content of wheat caryopses, leading to the increase in flour sedimentation value, wet gluten content, dry gluten content, and bread volume (Kimball et al., 2001; Ozturk & Aydin, 2004; Houshmand et al., 2005).

Wheat root system is the main place for water and nutrient absorption. Its morphological and physiological characteristics have a close relationship with the growth and development of wheat aboveground part, especially caryopses, thus affecting the formation of wheat yield and quality (Ober et al., 2021). The absorption of water by wheat roots is not only restricted by physical and mechanical properties of soil, but also affected by root growth and metabolism. In addition, the occurrence, development and physiological changes of root system depend on the moisture status of soil to some extent. Therefore, wheat root system and soil moisture are interdependent and restrict each other. Under drought stress, root number, root length, root specific surface area, root-shoot ratio, root growth potential, root water potential, and diameter of root vessel in wheat all changed significantly (Végh, 1991; Awad et al., 2018). Besides root morphological indexes, drought stress also had corresponding effects on some physiological indicators of roots, such as root bleeding sap, root respiration rate, root plasma membrane permeability, membrane lipid peroxidation level, protective enzymes activity and others (Kaul, 2011).

There are many morphological and physiological studies on wheat root system under drought stress at home and abroad and they are mainly concentrated in germination and seedling stages of wheat. However, researches on root system at later stage of wheat growth (booting stage and filling stage) is very weak. Very little information is available concerning structural characteristics of wheat roots under drought stress and the relationship between root morphology and caryopsis enrichment remains to be elucidated. In this study, Yangmai 13 (YM13) and Yannong 19 (YN19) were used as materials and subjected to drought stress from flowering to caryopsis maturity stage. The changes in morphological and structural characteristics of root and caryopsis development in wheat under drought stress were investigated by morphological observation and resin slicing. The relationship between root morphology and endosperm enrichment was also revealed. The results can not only enrich the cytological research of wheat roots, but also provide theoretical basis for root system architecture optimization, water-saving cultivation and quality improvement of wheat.

Materials & Methods

Plant materials and drought treatment

The wheat cultivars selected in this study were YM13 and YN19, which were provided by the Institute of Agricultural Science of the Lixiahe District in Jiangsu Province. These two wheat cultivars are widely cultivated in the region of Yangtze River with different degrees of response to drought stress. Seeds were sown in plastic pots (20 seeds per pot), which were placed in rainproof shelters in the experimental field of Key Laboratory of Crop Genetics and Physiology in Yangzhou University from October 2018 to May 2019. The soil is sandy loam soil with an organic matter content of 2.45%, nitrogen content of 106 mg/kg, phosphorus content of 33.8 mg/kg, and available potassium content of 66.4 mg/kg. Seedlings were thinned to six plants per pot two weeks after sowing. During flowering period, two individual florets at the base of central ears were marked with a marker pen and the plants were tagged with anthesis date.

Drought treatment was carried out by strictly controlling watering during the stage from flowering to caryopsis maturity referring to previous researches (Yang et al., 2004; Yu et al., 2015b). A minupressure soil hygrometer was inserted to measure the water potential at a depth of 20 cm in the soil. The water potential of control condition and drought stress were controlled at −20 and −60 kPa, respectively. Each treatment group contained 20 pots.

Morphological observation of plants, roots and caryopses

The soil in the plot was washed with water at 30 days post anthesis (DPA) and the roots were cleaned with an agricultural compression sprayer to observe the morphology by taking photographs. The root length and plant height were also measured. Then, the roots and aboveground parts of the plant were separated and baked in an oven at 42 °C to constant weight. The dry weight was determined and the root-shoot ratio was calculated. In addition, fresh caryopses at 10, 20 and 30 DPA were collected to determine the fresh weight and then dried in the oven at 42 °C to determine the dry weight. Wheat spikes were harvested at mature stage and the spike length, spike weight, grains number per spike and 1,000-grains weight were measured. Six wheat plants were selected as replicates for the above indicators determination.

Histological observation of roots and caryopses

Wheat root and caryopsis samples were collected as previously described in Li et al. (2020). The root samples were the segments of secondary root which were 2 cm from the stem axis base and the caryopsis samples selected were in the same position of the middle of main spike. The steps of sample resin slicing and observation referred to the method of Chen et al. (2017). Specifically, the samples were cut transversely into 2 mm slices from the middle with a razor blade and quickly soaked into 2.5% glutaraldehyde fixative solution at 4 °C for 48 h. The fixed samples were subsequently rinsed thrice (10 min each time) with phosphate buffer solution and dehydrated by gradient ethanol, followed by the replacement of propylene oxide. Then, the samples were infiltrated and embedded using Spurr resin and polymerized in an oven at 70 °C for 12 h. Afterwards, the ultramicrotome (Ultracut R, Leica, Germany) was used to cut the samples into 1 µm slices and stained with 0.5% methyl violet. The slices were observed and photographed under a light microscope (DMLS, Leica, Germany) equipped with a CCD camera (Truechrome II, Truechrome, China). Each treatment contained three replicated samples and each root and caryopsis sample was from different plants of different pots.

Structural characteristics analysis of endosperm enrichment

Image-Pro Plus 6.0 and Photoshop CC 2017 were used to analyze the relative areas of amyloplast and protein body in endosperm based on microphotographs, according to the method previously described (Yu et al., 2015a; Li et al., 2020). Specifically, the central endosperm region was photographed at 200 times magnification. The amyloplast, protein body and corresponding endosperm cell in the microphotograph were colored using Photoshop and the areas of colored regions were measured using Image-Pro Plus. The areas of amyloplast and protein body respectively divided the area of their corresponding endosperm cells were defined as the relative areas of amyloplast and protein body. Three caryopsis samples from different plants were selected as replicates for each treatment and ten micrographs were analyzed for each sample.

Statistical analysis

The experimental data were recorded using Microsoft Excel 2016. The significance analysis (Duncan method, p < 0.05) and Pearson correlation coefficient (p < 0.05) were conducted using SPSS 19.0 software. The figures were produced using Photoshop and Origin 9.1 software.

Results

Plant growth of wheat under drought stress

The morphological indexes of wheat plants under drought stress are shown in Table 1. Drought stress significantly reduced the plant height of YM13 and the root length, root dry weight and aboveground parts dry weight of YM13 and YN19, but had no significant effect on the plant height of YN19. At the same time, YM13 showed higher reduction degree in the root length and plant height under drought stress than YN19. While YN19 showed higher reduction degree in the root dry weight and aboveground parts dry weight than YM13. Under drought stress, the root-shoot ratio of YM13 and YN19 increased by 17.65% and 8.33%, respectively. In addition, drought stress significantly decreased the spike length, spike weight, grains number per spike, and 1,000-grains weight of YM13 and YN19. The range of decrease in the spike length and 1,000-grains weight was higher in YM13 than that in YN19 (Table 1).

Table 1 Plant morphology indexes and spike traits of wheat under drought stress.

Cultivars	Treatment	Root length (cm)	Plant height (cm)	Dry weight of roots (g)	Dry weight of aboveground parts (g)	Root-shoot ratio	Spike length (cm)	Spike weight (g)	Grains number per spike	1,000-grains weight (g)	
YM13	CC	36.93  ± 1.77a	63.77  ± 0.83a	3.47  ± 0.47a	10.27  ± 0.92a	0.34  ± 0.08b	8.14  ± 0.37a	2.48  ± 0.11a	37.67  ± 1.99a	36.80  ± 5.19a	
	DS	31.93  ± 1.28b	60.27  ± 1.41b	2.81  ± 0.45b	7.09  ± 1.04b	0.40  ± 0.03a	7.06  ± 0.32b	1.67  ± 0.10b	34.83  ± 2.59b	33.22  ± 4.06b	
YN19	CC	28.27  ± 1.65c	56.23  ± 1.84c	2.45  ± 0.53b	6.85  ± 0.55c	0.36  ± 0.06b	7.31  ± 0.16b	1.68  ± 0.17b	30.56  ± 1.57c	33.41  ± 0.85b	
	DS	25.70  ± 1.84d	54.67  ± 1.59c	1.52  ± 0.42c	3.85  ± 0.04d	0.39  ± 0.10a	6.36  ± 0.06c	1.04  ± 0.05c	27.17  ± 1.54d	31.70  ± 1.61c	
Notes.

Data are shown as mean ± standard deviation, n = 6. Different lowercase in the same column indicate significant difference (p < 0.05).

CC control condition

DS drought stress.

Morphological structure of wheat roots under drought stress

The root morphology of wheat in the whole pot was observed and it was found that the root system of YM13 was smaller and shorter while the root system of YN19 did not change significantly under drought stress (Fig. 1A). From the microstructure of root cross section, the cortex, vascular cylinder, vessel, and phloem can be observed. Under drought stress, the area of vascular cylinder became larger and more xylem vessels were differentiated and tightly arranged in the cross section of YM13 roots (Figs. 1B and 1C). However, the structure of root cross section in YN19 did not change significantly under drought stress and there was no significant difference in the area of vascular cylinder and the number of xylem vessels compared with control (Figs. 1D and 1E).

Figure 1 Morphology and structure of wheat roots under drought stress.

(A) Morphology of wheat roots. (B–C) Microstructure of YM13 root cross section under control condition and drought stress. (D–E) Microstructure of YN19 root cross section under control condition and drought stress. CC, control condition; Co, cortex; DS, drought stress; Ph, phloem; VC, vascular cylinder; XV, xylem vessel. Scale bars: (A) 8 cm, (B–E) 50 µm.

Enrichment of wheat caryopses under drought stress

The changes in the fresh and dry weight of wheat caryopses under drought stress are shown in Fig. 2. During the development of caryopses, the fresh weight first increased and then decreased, while the dry weight showed a continuous trend of increase. Under drought stress, the fresh weight of caryopses in YM13 and YN19 significantly decreased during the whole development process (Fig. 2A). At 10 DPA, the dry weight of caryopses in YM13 and YN19 did not change significantly under drought stress, while drought stress significantly reduced the dry weight at 20 and 30 DPA (Fig. 2B). In addition, YM13 presented greater reduction in the fresh and dry weight of caryopses under drought stress than YN19.

Figure 2 Changes in caryopsis weight and endosperm enrichment in wheat under drought stress.

(A) Fresh weight of caryopses. (B) Dry weight of caryopses. (C) Ratio of amyloplast area to endosperm area. (D) Ratio of protein body area to endosperm area. CC, control condition; DS, drought stress. Different lowercase above the histogram indicate significant difference (p < 0.05).

Substance accumulation characteristics in wheat endosperm was observed at 10, 20 and 30 DPA, and micrographs are shown in Fig. 3. At the same time, the relative areas of amyloplast and protein body in endosperm cells were calculated using Image-Pro Plus software (Figs. 2C and 2D). At 10 DPA, there is some starch accumulating in endosperm cells and distributing on the edge of cells. For two wheat cultivars, drought stress significantly decreased the accumulation of amyloplast in endosperm and YM13 showed greater decrease than YN19 (Fig. 2C, Figs. 3A–3D). A small amount of protein body was also observed in endosperm cells, but the change of protein body accumulation under drought stress was not obvious. Moreover, there was no significant difference in the response degree to drought between two wheat cultivars (Fig. 2D).

Figure 3 Microstructure of endosperm in wheat caryopses under drought stress.

(A–D) Microstructure of endosperm at 10 DPA. (E–H) Microstructure of endosperm at 20 DPA. (I–L) Microstructure of endosperm at 30 DPA. Am, amyloplast; CC, control condition; DS, drought stress; PB, protein body. Scale bars: 40 µm.

At 20 DPA, the endosperm of YM13 was further enriched and the volume and amount of amyloplast increased. The accumulation and aggregation of some small endosperm protein body was also clearly observed in YM13 (Figs. 3E and 3F). Under drought stress, the relative area of amyloplast significantly decreased, while the relative area of protein body significantly increased in YM13 (Figs. 2C and 2D). For YN19, the endosperm was not as full as YM13 and there were large gaps between amyloplast. Some protein aggregations coalesced by small protein body were also observed in endosperm (Figs. 3G and 3H). Meanwhile, drought stress significantly decreased amyloplast accumulation but increased protein body accumulation of endosperm in YN19 (Figs. 2C and 2D). For the relative area of endosperm amyloplast, the reduction degree under drought stress was higher in YM13 than that in YN19 (29.03% vs. 16.47%). For the relative area of endosperm protein body, the augment degree under drought stress was higher in YM13 than that in YN19 (22.88% vs. 8.96%).

At 30 DPA, the endosperm of two wheat cultivars was entirely enriched. The amyloplast were squeezed each other and the protein body with enlarged volume accumulated in the gaps of amyloplast (Figs. 3I–3L). Drought stress significantly reduced the relative area of endosperm amyloplast and increased the relative area of endosperm protein body in both YM13 and YN19 (Figs. 2C and 2D). However, the decrease of amyloplast accumulation and the increase of protein body accumulation were more responsive to drought stress in YM13 than those in YN19.

The above results showed that drought stress could affect the substance accumulation in wheat caryopses by reducing starch accumulation and increase protein accumulation in endosperm, thus resulting in the decrease of caryopsis weight. However, the impact of drought stress on YM13 was greater than that on YN19.

Relationship of root morphological structure and endosperm enrichment in wheat under drought stress

In order to investigate the relationship between root morphological structure and endosperm substance accumulation, the correlation analysis of roots and aboveground parts traits was carried out. The root length was negatively correlated with the plant height, aboveground parts dry weight and 1,000-grains weight, and positively correlated with the spike weight, relative areas of amyloplast and protein body. The root length had a significantly strong correlation with the plant height and relative area of protein body under control condition, while the root length had a significantly strong correlation with the relative areas of amyloplast and protein body under drought stress. This indicated that drought stress reduced the correlation between the root length and the plant height, and increases the correlation between the root length and the relative area of amyloplast (Table 2). Except for the relative area of protein body, the root dry weight was positively correlated with other aboveground parts traits. Among them, the root dry weight was closely related to the plant height and aboveground parts dry weight with a significance under control and drought stress. However, drought stress reduced the correlation between the root dry weight and the 1,000-grain weight (Table 2). In addition, the root-shoot ratio was negatively correlated with other traits except the 1,000-grains weight. Under control condition, the root-shoot ratio had a significantly strong correlation with the aboveground parts dry weight, spike weight and relative area of amyloplast. While, drought stress decreased correlation coefficients of the root-shoot ratio with the aboveground parts dry weight and spike weight, and enhanced the correlation coefficients of the root-shoot ratio with the relative areas of amyloplast and protein body (Table 2). The above results indicated that the root length was strongly correlated with the relative areas of amyloplast and protein body, and drought stress increased correlation coefficients between them. Meanwhile, the root dry weight had a great correlation with the plant height and aboveground parts dry weight.

Table 2 Correlation coefficients between traits of roots and aboveground parts in wheat under drought stress.

Indexes	Plant height	Dry weight of aboveground parts	Spike weight	1,000-grains weight	Relative area of amyloplast	Relative area of protein body	
Root length	CC	−0.78∗	−0.33	0.37	−0.54	0.51	0.70∗	
DS	−0.14	−0.57	0.67	−0.45	0.79∗	0.78∗	
Dry weight of roots	CC	0.85∗	0.99∗	0.48	0.58∗	0.41	−0.22	
DS	0.89∗	0.98∗	0.29	0.19	0.29	−0.03	
Root-shoot ratio	CC	−0.43	−0.74∗	−0.61∗	0.27	−0.85∗	−0.58	
DS	−0.15	−0.39	−0.33	0.57	−0.86∗	−0.75∗	
Notes.

Each correlation coefficient is calculated from six values. Asterisks in the upper right corner of the number indicate that the correlation coefficient is significant (p < 0.05).

CC control condition

DS drought stress.

Discussion

The growth of root system is related to the growth period and genotype of plants and is affected by environmental factors such as drought (Rellán-Álvarez, Lobet & Dinneny, 2016). It is generally believed that roots absorb water from the soil first to meet the need of its own growth and development. Thus, the damage degree of roots is lighter than that of shoots when exposed to water stress, leading to the increase of root-shoot ratio. Under severe water shortage, the elongation and growth of wheat roots were severely hindered and the dry matter of roots was significantly reduced (Xue et al., 2003). In this study, drought stress decreased the length and dry weight of wheat roots and affected the growth of aboveground parts, resulting in a decline in the plant height and aboveground parts dry weight. Ultimately, the yield of wheat spikes was also decreased. The root vascular bundle is a channel for the transportation of water and inorganic salts, which is essential to cope with water changes even deficits. In this study, the cross section structure of wheat roots was observed and it was found that the area of vascular cylinder was larger with denser distribution of xylem vessels under drought stress. This phenomenon might be related to water flow conductivity. The radial and axial resistance are the main factors affecting the water flow conductivity of the root system (Vadez, 2014). Large vessels will appear the embolism formed by bubbles, thus the radial and axial hydraulic resistance will increase (Niu et al., 2016). Thereby, the root water flow conductivity is reduced and the root water absorption is limited, which is not conducive to plant growth under drought stress (Bartlett et al., 2016). While, the small and narrow xylem vessels could transport water more efficiently, which was similar to the results of previous researches (Thorsten & Wieland, 2011; Comas et al., 2013).

Wheat roots is the organ that absorbs water and senses environmental changes in the soil. Many root characteristics are related to drought tolerance, such as root depth, angle, density, surface area and xylem diameter. They are all root characteristics that affect drought tolerance of wheat and is closely related to the growth and yield of wheat. Previous studies have investigated the relationship between root architectural traits and yield components in wheat seedling, and it is found that seminal root number and total root length were both positively associated with aboveground biomass, grains per spike, and grain yield (Xie et al., 2017). This is consistent with the results in this study that the root dry weight was highly correlated with the plant height and aboveground parts dry weight. In the study of Li et al. (2019), the root depth of wheat was significantly negatively correlated with canopy temperature and significantly positively correlated with the yield per plant. But the root depth had no significant correlation with the plant height and root dry weight. This reflected the complexity of root architecture, that was the diversity of root dry matter distribution in different soil layers. In present study, the relationship between root growth and caryopsis enrichment in wheat was analyzed, and it was found that the root length had a great correlation with the relative area of amyloplast and protein body and drought stress increased the correlation coefficients between them, which explained the relationship between root development and quality formation in wheat to a certain extent. In general, the root architecture is of great importance to yield, quality and stress tolerance of wheat. Optimizing root architecture and investigating the balance relationship between root morphological structure and aboveground agronomic traits are powerful means to promote variety improvement and improve drought resistance in wheat.

Conclusions

In this study, the morphological structure of roots and characteristics of endosperm enrichment in wheat under drought stress were investigated. Drought stress significantly reduced the root length, plant height, root dry weight and aboveground parts dry weight but increased the root-shoot ratio. The spike length, spike weight, grains number per spike and 1,000-grains weight of mature wheat also significantly declined under drought stress. Meanwhile, the cross section structure of roots was changed under drought stress with the enlargement of vascular cylinder and dense distribution of xylem vessels. In additional, drought stress affected the substance enrichment in wheat caryopses with a decrease in starch accumulation but an increase in protein accumulation of endosperm. Correlation analysis suggested that the root length was closely correlated with the relative areas of amyloplast and protein body, and drought stress increased the correlation coefficient. While the root dry weight had a significantly positive correlation with the plant height and aboveground parts dry weight. The results can provide insight into root growth and caryopsis development of wheat under drought stress and guide root architecture optimization and quality improvement in wheat.

Supplemental Information

Supplemental Information 1 Raw data of measurements

Click here for additional data file.

Additional Information and Declarations

Competing Interests

Author Contributions

Data Availability

The authors declare there are no competing interests.

Xinyu Chen performed the experiments, analyzed the data, prepared figures and/or tables, authored or reviewed drafts of the paper, and approved the final draft.

Yu Zhu, Yuan Ding and Rumo Pan performed the experiments, prepared figures and/or tables, and approved the final draft.

Wenyuan Shen performed the experiments, analyzed the data, prepared figures and/or tables, and approved the final draft.

Xurun Yu and Fei Xiong conceived and designed the experiments, authored or reviewed drafts of the paper, and approved the final draft.

The following information was supplied regarding data availability:

The raw measurements are available as a Supplementary File.

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
