# Peer review of "The relationship between characteristics of root morphology and grain filling in wheat under drought stress"

_PeerJ, doi:10.7717/peerj.12015_

## Round 0.1 · original submission · Minor Revisions

Please revise your manuscript according to the comments from reviewers.

Reviewer 1 ·

Basic reporting

In Table 2, the p value of asterisks was missed.

Experimental design

In the "M&M' section, the author performed the significance analysis by using LSD method, Line144-145. As well known, LSD method is always used to detect the difference between CKand its treatment. For example, you have 4 groups, CK, treatment 1, treatment 2, treatment 3. LSD could detect the difference between CK and treatment 1,CK and treatment 2, CK and treatment 3, but not well used to tell wether treatment 1,2,3 have significant difference. We suggest the author use "Duncan method" or other methods.
Also, the author should give the p value they use (0.05 or 0.01 or other value) while calculating the Pearson correlation coefficient Line145.

Validity of the findings

no comment.

Additional comments

This article analyzed the relationship ship between root morphology and grain filling under drought stress in wheat. the experiment design is reasonable. The results can provide theoretical basis for root architecture optimization, watersaving and high-yield cultivation and quality improvement in wheat. And can be accepted after minor revise.

Reviewer 2 ·

Basic reporting

This manuscript described the morphological and structural characteristics of root morphology of root growth and caryopsis enrichment, along with their relationships, in wheat under drought stress. The data are enough to lead their conclusion and the results can provide insights into root architecture optimization and quality improvement in wheat.

Experimental design

well

Validity of the findings

no commet

Additional comments

This manuscript described the morphological and structural characteristics of root morphology of root growth and caryopsis enrichment, along with their relationships, in wheat under drought stress. The data are enough to lead their conclusion and the results can provide insights into root architecture optimization and quality improvement in wheat. Revision is required before acceptance for publication and the questions are listed as below.
1. The abstract need to be more informative. The cultivars of wheat used for the experiment and the specific value of changes in wheat traits under drought stress should be mentioned in abstract.
2. In this study, Yangmai 13 and Yannong 19 were used as materials. Please explain why select these two cultivars and describe these two cultivars in more details.
3. In this study, the control condition and drought stress were set at two different water potential (-20 and -60 kPa). Why use this treatment and is there any references supporting this?
4. Line 113, 116, 124, the font of ℃ seems to be different with the style of the whole manuscript. Please check it.
5. Line 140-141, please explain the term “relative areas of amyloplast and protein body” that was mentioned in materials and methods section in more details.
6. The botanical names in references section should be standard. Please check it.
7. In Table 2, the p-values should be given alongside the correlation coefficient to indicate significance.
8. In Figure 1, the scale bar of (A) is 8 mm in figure legend while is 8 cm in the figure. Which one is correct? Please confirm it.

---

## Round 0.2 · accepted · Accept

Your manuscript is acceptable based on the recommendation from the reviewers.

Reviewer 1 ·

Basic reporting

no comment

Experimental design

no comment

Validity of the findings

no comment

Additional comments

This article analyzed the relationship ship between root morphology and grain filling under drought stress in wheat. the experiment design is reasonable. The results can provide theoretical basis for root architecture optimization, watersaving and high-yield cultivation and quality improvement in wheat. And can be accepted.

Reviewer 2 ·

Basic reporting

good

Experimental design

good

Validity of the findings

good